# Profiling Risk Factors for Household and Community Spatiotemporal Clusters of Q Fever Notifications in Queensland between 2002 and 2017

**DOI:** 10.3390/pathogens11080830

**Published:** 2022-07-25

**Authors:** Tatiana Proboste, Nicholas J. Clark, Sarah Tozer, Caitlin Wood, Stephen B. Lambert, Ricardo J. Soares Magalhães

**Affiliations:** 1UQ Spatial Epidemiology Laboratory, School of Veterinary Science, University of Queensland, Gatton, QLD 4343, Australia; t.probosteibertti@uq.edu.au (T.P.); n.clark@uq.edu.au (N.J.C.); 2Queensland Maternal and Perinatal Quality Council, Queensland Department of Health, Herston, QLD 4029, Australia; sarah.tozer@health.qld.gov.au; 3School of Veterinary Science, University of Queensland, Gatton, QLD 4343, Australia; caitlin_mcguckin@hotmail.com; 4National Centre for Immunisation Research and Surveillance, Westmead, NSW 2145, Australia; sblambert@uq.edu.au; 5UQ Centre for Clinical Research, University of Queensland, Herston, QLD 4006, Australia; 6Children’s Health and Research Centre, Children’s Health and Environment Program, University of Queensland, South Brisbane, QLD 4101, Australia

**Keywords:** *Coxiella burnetii*, Q fever, cluster analysis, Queensland, spatiotemporal analysis, Q fever notification

## Abstract

Q fever, caused by the bacterium *Coxiella burnetii*, is an important zoonotic disease worldwide. Australia has one of the highest reported incidences and seroprevalence of Q fever, and communities in the state of Queensland are at highest risk of exposure. Despite Australia’s Q fever vaccination programs, the number of reported Q fever cases has remained stable for the last few years. The extent to which Q fever notifications cluster in circumscribed communities is not well understood. This study aimed to retrospectively explore and identify the spatiotemporal variation in Q fever household and community clusters in Queensland reported during 2002 to 2017, and quantify potential within cluster drivers. We used Q fever notification data held in the Queensland Notifiable Conditions System to explore the geographical clustering patterns of Q fever incidence, and identified and estimated community Q fever spatiotemporal clusters using SatScan, Boston, MA, USA. The association between Q fever household and community clusters, and demographic and socioeconomic characteristics was explored using the chi-squared statistical test and logistic regression analysis. From the total 2175 Q fever notifications included in our analysis, we found 356 Q fever hotspots at a mesh-block level. We identified that 8.2% of Q fever notifications belonged to a spatiotemporal cluster. Within the spatiotemporal Q fever clusters, we found 44 (61%) representing household clusters and 20 (27.8%) were statistically significant with an average cluster size of 3 km radius. Our multivariable model shows statistical differences between cases belonging to clusters in comparison with cases outside clusters based on the type of reported exposure. In conclusion, our results demonstrate that clusters of Q fever notifications are temporally stable and geographically circumscribed, indicating a persistent common exposure. Furthermore, within individuals in household and community clusters, abattoir exposure (a traditional occupational exposure) was rarely reported by individuals.

## 1. Introduction

Q fever, caused by the bacterium *Coxiella burnetii*, is an important zoonotic disease worldwide. In humans, the bacterium can cause a range of disease patterns, including asymptomatic infection, mild influenza-like symptoms, through to chronic manifestations. Approximately 10–15% of acute cases progress to a chronic fatigue-like state labelled post-Q fever fatigue syndrome [1]. In ruminants, the bacteria causes coxiellosis, which affects reproductive performance, particularly in small ruminant species, presenting as disorders such as abortion and infertility [2].

Human exposure to *C. burnetii* in Australia is widespread, with one study suggesting that 1 in 20 Australians have evidence of neutralising antibodies [3]. Seroprevalence in adolescents shows that Q fever is an ongoing public health issue [3]. Queensland, the most northeast-most state in Australia, is home for 19.6% of the national population but reports 43.1% of national notifications [4] and has the highest average annual Q fever notification rate at 6.3 per 100,000 population per annum [5]. These figures are likely to be moderate underestimates, due to the failure to detect asymptomatic infections. A recent study identified that 89% of blood donors that showed previous exposure to *C. burnetii* have never had a Q fever diagnosis [6].

The cornerstone of Australia’s Q fever control includes vaccination and education programs focused on people identified as being at higher risk, such as workers in close contact with animals, particularly those working in abattoirs, farms, and veterinary clinics [7]. However, as the National Q Fever Management Program results in substantial improvement in the burden of Q fever in these sectors, there is increasing evidence of infection of other sectors, including those residing in urban and suburban areas [8]. Moreover, given continued urbanisation in traditional farming areas, there is rising concern over the potential for airborne spread of Q fever to communities neighbouring animal industries and processing facilities [9].

In our previous work, we detailed that reported animal exposure patterns in Queensland differ markedly depending on where cases live in the state [10], pre-empting the need for a deeper investigation into whether cases exhibit spatiotemporal clustering and how demographic and contextual profiles of the cases vary across the state. There is a significant gap in our understanding of exposure pathways to *C. burnetii* within high-risk communities, and of the complexities of Q fever epidemiology to help design measures aiming at the prevention of *C. burnetii* exposure [10].

Q fever represents a diagnostic challenge, particularly in those without a history of occupational exposure, hence is considered an underdiagnosed disease with the true infection rate within the community likely higher than the notification rate [5,11]. Household-community clusters, which to date have not been adequately studied in Australia, represent an opportunity to better understand the complex epidemiology of Q fever transmission locally by examining differences between and within household and community clusters. However, to determine the approach to investigate household and community clusters, it is essential to understand how often these clusters occur and their relative location to known geographical areas of Q fever notifications and the differences in reported exposures between individuals in household and community clusters and other Q fever cases.

This study aimed to retrospectively explore the spatiotemporal clustering patterns of Q fever notifications in Queensland between 2002 and 2017, identify household and community clusters and compare epidemiological features of cases within community and household clusters to cases from those outside of clusters.

## 2. Results

### 2.1. Geographical Clustering of Q Fever Incidence at the Mesh-Block Level

A total of 2175 out of 3233 records had a valid home address within Queensland borders during the period between 2002 and 2017. The data excluded from the analysis corresponded to 78% (*n* = 827) of records without an address, concentrated in 2002 (*n* = 164) and 2003 (*n* = 90), with the proportion of notifications with a missing address decreasing across the years. For 231 records, the address was not recognised within OpenStreetMap.

The spatial analysis of all Q fever notified cases in Queensland indicated significant clustering in that the overall Moran’s I estimate was 0.033 (Z-value: 15.7818; *p*-value: 0.002). For 2008, 2009 and 2016, the estimated Moran’s I value was negative, indicating no significant clustering for those years. Across the years with a positive Moran’s I value, the *p*-value varied between 0.002 (2015) and 0.06 (2007) (Table 1).

A total of 356 Q fever incidence hotspots (i.e., mesh blocks classified as high-high by LISA analysis) were primarily distributed in South East Queensland, close to the border with New South Wales, and 10 mesh blocks were classified as high-high for more than one year across the study period with 4 mesh blocks classified as high-high across five years (Figure 1). Our annual LISA analysis indicated that in 2003 we found a high number of mesh blocks classified as high-high (*n* = 18), which decreased during subsequent years, to increase again in 2013 (*n* = 13) and 2014 (*n* = 26) (Table 1).

### 2.2. Spatiotemporal Variation in Q Fever Household and Community Clusters

The location of household and community clusters identified by space–time scan statistics for the whole period shows clusters primarily in southeast Queensland and on the coast of Townsville, in the state’s northeast. The annual number of cases per 100,000 people was 2.9. From the total Q fever cases reported, 8.2% (*n* = 179) belonged to a spatiotemporal cluster. We identified 72 spatiotemporal clusters across the study period between 2002 and 2017, using spatiotemporal scan statistics. From the 72 spatiotemporal clusters identified, 28 belonged to community clusters and 44 belonged to household clusters (Appendix A). The average community cluster size was 3 km radius. The model revealed 20 significant clusters (Table 2), with the largest number of cases in the Townsville cluster, with eight observed cases and a radius of 9.66 km.

The time frame for Townsville’s cluster was from February to March 2012 and carried a relative risk (RR) of 184.39 and a log-likelihood ratio (LLR) of 33.77. In addition, the 19 remaining clusters identified have a relative risk from 868 to 499.74.

### 2.3. Profile of Exposures of Q Fever Cases within Household and Community Clusters

Our analysis of the exposure responses of Q fever cases between household and community clusters detected by space–time analysis is summarized in Figure 2, and represents all different types of exposure reported by 179 cluster-associated cases. A total of 50% of recorded cases answered positively to living or working within 300 m of bush, followed by exposure to paddock dust (46%), and being exposed to livestock transport, and assisting/observing animal birth (33%). On the other hand, only 3% of recorded cases reported abattoir exposure and 1% reported working in the grounds of an abattoir.

### 2.4. Factors Associated with the Probability of Belonging to a Household or Community Q Fever Cluster

We analysed the reported exposure profile for each cluster type (community, household, or the combination of both) and cases reported outside a cluster. Our results indicate that the reported exposure profiles of Q fever notified cases within a cluster differed significantly from those of Q fever notified cases outside clusters. Factors independently associated with belonging to a Q fever household or community cluster included having contact with an infected person (*p* ≤ 0.001), which was statistically significant for all groups (household clusters only, community clusters only, and the combination of both cluster types). Assisting/observing animal birth (*p* ≤ 0.001) was statistically significant for community and household clusters as well as laundering clothes of an animal worker (*p* ≤ 0.001) and living on a farm (*p* ≤ 0.001) (Table 3).

In the Generalise Additive Model (GAM), cases belonging to a community and household cluster were more likely to report being in contact with an infected person in the one month prior to disease notification (*p* ≤ 0.001). Cases belonging to a household and community cluster were also more likely to have reported assisting with or observing an animal birth (*p* = 0.036) than cases reported outside a cluster (Table 4).

## 3. Discussion

In this study, we have identified significant overall geographical clustering in Q fever notifications in Queensland for the period of 2002 to 2017, suggesting common pathways of exposure to *C. burnetii* in vulnerable communities. Our results found clustering for 11 out of the 16 years analysed, and nonsignificant clustering was correlated to periods when Q fever notification incidence was relatively low. Results from a previous study indicated that during 2007, 2008, and 2009 there was a sharp decrease in the Q fever notification rate in Queensland, followed by an increase in 2010 [4] which correlates with our clustering results for the 2007–2009 period. We found the highest Moran’s I value (0.02) in 2015, which corresponded with the second-largest peak of Q fever notifications in Queensland in the past 20-years [4]. Our study extends previous research in that we were able to identify Q fever incidence hotspots in communities in the southeast interior of the state as well as the northern tropical region. In previous work, we [4] described higher notification rates (per 100,000 population) in the Mareeba district, located in Far North Queensland, but while we did not identify statistically significant clusters in that area, we identified a significantly higher rate of Q fever notifications around the Townsville region. Moreover, our results demonstrate that clusters of Q fever notifications are temporally stable and geographically circumscribed, which may be an indicator of the existence of a persistent common exposure. Furthermore, individuals in household and community clusters do not seem to report abattoir exposure as the main exposure pathway, a traditional occupational risk group currently targeted for Q fever vaccination.

Our spatiotemporal analyses identified a total of 72 spatiotemporal Q fever clusters in Queensland between 2002 and 2017, 20 of which were statistically significant spatiotemporal clusters across Queensland. Our results indicate that Q fever clusters are an important component of Queensland Q fever notifications, as 8.2% of cases are generally associated with a spatiotemporal cluster. The average Q fever community spatiotemporal cluster was estimated to be of 3 km radius, which is in line with existing evidence indicating that the risk of *C. burnetii* infection is higher within 5 km of a contaminated source in rural areas [9]. Studies conducted with data collected during Q fever outbreaks indicate that the risk of infection is high in the direct vicinity of a source, decaying very rapidly after that. For example, the outbreak in Germany in 2005 had an association between risk of infection with Q fever and living close to a meadow with *C. burnetii*-infected sheep grazing and lambing. The attack rate during this outbreak dropped from 11.8% within 50 m to 1.3% at 350–400 m [12]. Our results also demonstrated that community clusters were located across the whole state, with the majority located in southeast Queensland, the western clusters across Murweh, Blackall Tambo Barcaldine regions, and the northern cluster located in Townsville region. The Townville cluster corresponded to the biggest cluster identified in our analysis, with a radius of 9.6 km. The large size of the Townsville cluster is consistent with evidence that *C. burnetii* can travel long distances, up to 18 km, by strong winds [13]. Aerosol dispersal of *C. burnetii* via wind has been associated with outbreaks in France, Germany, Netherlands, and UK [9,12,13,14,15,16], and in outbreak conditions it has been reported that Q fever cases can cover approximately 10 km^2^ areas [17]. However, other factors such as the average size of the mesh block in the Townsville area (larger than in the southeast region) could have an effect on the size of the clusters. While community clusters were located across the whole state, household clusters analysed in our study were mainly concentrated in the southeast region. Household clusters, in which members from the same house were exposed to the bacteria without necessarily having an ‘at risk’ occupation, were identified in our spatiotemporal analysis. For example, only 3% of people identified as part of a household cluster reported abattoir exposure. This result is supported by an increasing number of Q fever reports that are not related to direct contact with animals [18]. Our results from the household cluster profile indicate there may be a role for expanding Q fever control measures to people and communities that do not necessarily fit the current ‘at risk’ list of occupations.

Despite the endemicity of Q fever in Australia, epidemiological studies on Q fever are generally missing information about the infection risk profile of communities with recurrent Q fever risk that could inform the evidence base for the existence of a putative source of infection [9]. Our findings indicate that Q fever notified cases belonging to Q fever spatiotemporal clusters (community, household, or the combination of both) are associated with particular modifiable exposures, compared to Q fever notified cases outside identified clusters. This result suggests that sociodemographic context within identified Q fever spatiotemporal needs be taken into consideration when designing health promotion and education strategies to reduce potential sources of *C. burnetii* exposure. Interestingly, we did not find differences in abattoir exposure between Q fever cases belonging to a cluster and those not belonging to a cluster. Exposures other than abattoir-related exposure are likely to distinguish Q fever cases in household and/or community clusters from other cases. Indeed, our results indicate that Q fever cases are more likely to belong to a family and community cluster if they assist animal birth [19,20,21] or have contact with an infected person. The univariable model also shows that those cases reporting contact with clothes worn by someone who worked with animals were more likely to belong to a cluster. This type of exposure has been previously reported in a small outbreak, with three laundry workers infected with Q fever [22]. Our results suggest that laundered clothes from animal workers are a potential risk source for Q fever clusters. Similarly, we identified that notifications that reported exposure to paddock dust were more likely to belong to a community or household cluster. This result is consistent with the importance of aerosol transmission in Q fever infections [9] due to the capacity of the bacteria to survive in the environment, with viable bacteria being recovered from soil up to 20 days after inoculation [23]. Cases reporting living or working within 300 m of bushland were also more likely to belong to a cluster. This may be an indicator that the environment is playing an important role in the maintenance of the bacteria that could drive the Q fever clusters. The exposure reported of being in contact with an infected person in the month prior to the disease onset is an expected outcome, as it aligns with the cluster definition used in this study, defined by a minimum of two cases in two months. In addition, this result demonstrates that Q fever reported cases within these clusters are familiar with Q fever, since they are likely to know someone who has had Q fever.

As with all observational studies, there are limitations in our work. First, the records of Q fever cases are not always complete. For this study, 827 cases had an incomplete address, and therefore were removed from the analysis, and more than half of the cases that belonged to a household cluster had no information about their place of work. Secondly, the limitation of the ScatScan analysis due to the lack of an autoregressive process to capture the temporal dependencies.

## 4. Materials and Methods

### 4.1. Data Sources and Management

Q fever is a notifiable condition in Queensland under the Public Health Act 2005 and its subordinate legislation [24,25]. Q fever notification records from 2002 to 2017 were obtained from the Notifiable Conditions System (NoCS) managed by the Communicable Disease Branch of Queensland Health. The Notifiable Conditions System compiles data from clinical information, with follow-up from select individual public health units (PHUs) via case reporting forms. From 2012 onwards, Q fever notified cases have been contacted by staff of associated PHU and asked to respond to additional follow-up questions using a Q fever case report forms to collect information. The information included for this analysis is based on reported exposures in the month prior to illness onset (Queensland Health). Records between 1 January 2002 to 31 December 2017 with complete home addresses were included in the analysis. All cases were geocoded at the street address using the package *tmaptool* [26] in R [27] and the ©OpenStreetMap contributors; records outside Queensland borders were removed from the analysis.

We used human population counts and demographic data in Queensland at the mesh-block statistical area, obtained from the ‘2074.0-Census of population and housing: Australia, 2016’ [28]. We used mesh-block divisions obtained from ASGS Ed 2016 digital boundaries in ESRI Shapefile format [28,29]. Isolated polygons such as islands were removed prior to the analysis.

To perform spatial analysis of Q fever incidence across Queensland, the mesh block was considered the spatial unit of analysis. Using the spatial join tool in ArcGIS Pro (version 2.7.0), we counted the number of Q fever notifications for each mesh block, and incidence was calculated by dividing the Q fever count per mesh block by the total population of the mesh block. When the population in a polygon was equal to 0 and Q fever records ≥1, we used the population from the nearest neighbour that had a recorded population.

The identification of space–time community clusters was performed by aggregating the geographical location of each case to the centroid of the mesh block and the population was also included as mesh-block level. Data management was conducted in ArcGIS Pro and the software R [27].

### 4.2. Exploration of Q Fever Notification Clustering Patterns in Queensland

We used the Moran’s I statistic to assess the extent of spatial clustering of annual Q fever incidence (i.e., observed cases per 100,000 population) at the mesh-block level for the period of 2002 to 2017. To explore the location of significant high-risk mesh blocks for Q fever incidence, we applied the Local Moran’s statistic, which is a Local Indicator of Spatial Association (LISA), to determine the spatial locations of the Q fever clusters in Queensland during each year. Using estimates of observed vs. expected incidence from the LISA analysis, each mesh block was categorized as a hotspot (high-high), coldspot (low-low) or as an outlier (high-low and low-high) [30]. A Z-score is generated by the Local Moran’s I statistic to determine the significance level of clusters. Surroundings with spatial clusters will be indicated by a high positive Z-score, and the presence of spatial outliers will be represented by a low negative Z-score. A pseudo *p*-value was calculated using 499 permutations; this value corresponds to a summary of the results from the null reference distribution that assumed notifications were randomly distributed across the study area [31]. We investigated hotspots’ mesh-block stability across the study period by spatially overlaying high-high mesh block for each year and selecting the mesh blocks that were categorised as hotpots in multiple years. Analyses were performed using GeoDa^TM^ software [32].

### 4.3. Identification of Q Fever Household and Community Clusters

We categorised clusters into three categorical levels: household clusters, community clusters, or a combination of both. When two or more cases were recorded from the same home address within a period of six months, we considered this as a household cluster. Household clusters were identified based on the data available on the notification report form, including records for which a georeference was not available, but a home address or a name of a property was provided.

To identify the presence of community clusters and respective cluster sizes, we explored the spatiotemporal pattern of Q fever notifications clusters by performing a spatial scan (SaTScan software, version 9.7). In this study, we defined a community cluster as two or more cases associated within a 10 km radius, as it has been previously described that infection risk is generally higher within 5 to 10 km from an infected source [9]. The time aggregation for this study was two months, based on the maximum incubation period reported of 60 days with a median incubation period of 18 days [33]. The geographical unit of analysis was the geographical centre of the mesh block; a mesh block corresponds to the smallest geographic region in the Australian Statistical Geography Standard. Therefore, the input data for this analysis consisted of (i) a Q fever case notification file, where all Q fever notified cases during the 2002–2017 period were summarised for each mesh block per month; (ii) a population file based on the Australian census 2016 by mesh block, and (iii) a geographic file, consisting of the centroid of each mesh block in Queensland.

We used a space–time scan analysis, which is defined by a cylindrical ‘window’, in which a circle represents the geographic base, and the time is represented by the height of the cylinder. Then, the cylinder is moved in space and time, creating overlapping cylinders, where each cylinder represents a possible cluster [34]. A retrospective space–time analysis and a discrete Poisson probability model were used to estimate relative risk. We scanned for areas with high and lower rates. A likelihood ratio test was used to compare the alternative hypothesis, that risk is higher within the window as compared to the outside, providing relative risk and *p*-values for each cluster [35]. The model was run using a standard Monte Carlo test with 9999 replications to generate a *p*-value. We compared the results from the space–time analysis with the household clusters identified previously, and we categorised each space–time cluster into household or community clusters. A community Q fever cluster corresponded to a space–time cluster that did not overlap with the previously identified household clusters based on home address.

### 4.4. Associations between Q Fever Clusters and Demographic and Socio-Economic Characteristics

For the purpose of this analysis, the household and community clusters included those identified by home address and those based on SatScan analysis, respectively. We extracted only the Q fever notifications (*n* = 179) that were part of a cluster identified by spatial scan. We investigated the reported at-risk exposure within one month prior to the notification date of Q fever. We explored: (i) differences between clusters; for example, if cases from the same cluster reported the same ‘at risk’ exposure, and (ii) we explored within each cluster the type of exposure reported. We used a Pearson’s chi-squared test to investigate differences between the type of exposure individuals reported within household, community, or household and community clusters, and individuals who did not belong to these clusters. We excluded from the analysis patient responses that were recorded as ‘unknown’, or that contained missing data. We use a penalised General Additive Model (GAM) to investigate whether belonging to a cluster was associated with exposure type. We excluded variables that were correlated providing similar information, and with threshold value for correlation coefficients > 0.5. For instance, we included abattoir exposure, while work inside abattoirs was excluded for the model. Similar with variables related to work with wool, we excluded work in a shearing shed, and work in wool processing. A total of 17 variables were included in the GAM, with smoothing penalty using *mgcv* package [36]. We performed automatic variable selection using a random effect basis with a double penalty approach to regularise coefficients toward zero. All statistical analyses were conducted in R [27].

## 5. Conclusions

This study provides a detailed spatiotemporal analysis of Q fever clusters in Queensland as well as insight into the different ‘at risk’ exposures described between cases belonging to clusters and cases outside clusters. We conclude that Q fever cluster communities identified in this study require an in-depth environmental risk assessment to help inform public health strategics to decrease their endemicity. Further analysis is needed to understand the epidemiology of *C. burnetii* within clusters, and to determine the main source of infections in these clusters.

## Figures and Tables

**Figure 1 pathogens-11-00830-f001:**
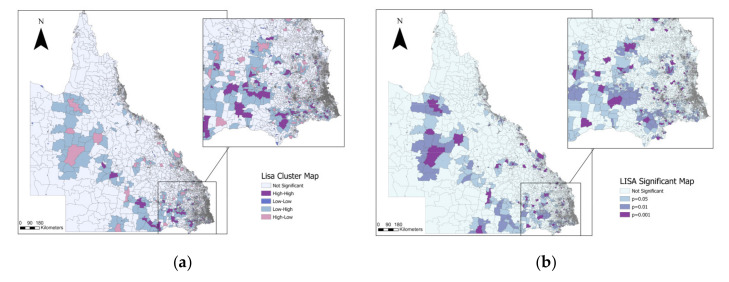
Local Indicators of Spatial Association (LISA) cluster map for the cumulative Q fever incidence in Queensland between 2002 and 2017. (**a**) Distribution of LISA clusters; (**b**) distribution of statistically significant LISA clusters.

**Figure 2 pathogens-11-00830-f002:**
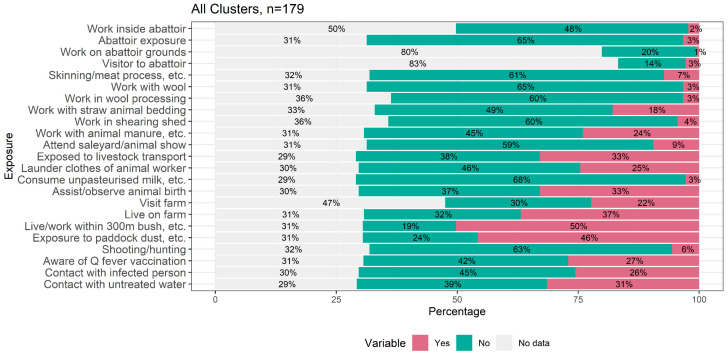
Percentage of responses for each Q fever exposure for all cluster cases.

**Table 1 pathogens-11-00830-t001:** Yearly spatial–temporal analysis of Q fever incidence in Queensland, 2002–2017.

Year	Moran’s I	Z-Score	*p*-Value	Number of HH LISA Clusters
2002	0.010	3.1425	0.006	17
2003	0.013	10.7822	0.004	18
2004	0.013	7.9687	0.004	6
2005	0.001	0.4871	0.052	4
2006	0.006	6.0461	0.01	7
2007	0.001	1.0464	0.064	6
2008	<−0.0001	−0.1928	0.342	2
2009	<−0.0001	−0.1908	0.411	0
2010	0.002	2.7421	0.018	13
2011	0.001	0.6546	0.054	8
2012	0.006	2.9126	0.028	7
2013	0.006	3.6136	0.012	13
2014	0.003	2.5158	0.026	26
2015	0.021	17.6224	0.002	20
2016	<0.0001	0.1294	0.162	6
2017	0.002	0.4815	0.048	8
All years	0.033	15.782	0.002	356

**Table 2 pathogens-11-00830-t002:** Significant Q fever household and community clusters using space–time analysis in Queensland between 2002 and 2017.

Cluster Locality	Radius (km)	Year	LLR	*p*-Value	Observed	Expected	RR	Population	Cluster
					Cases	Cases			
Paroo Shire	0.00	2002	51.46	0.00	6	0.00	14,453	86	H
Gympie Regional	0.00	2008	40.48	0.00	5	0.00	8920	118	H
Murweh Shire	1.35	2015	40.16	0.00	6	0.00	2196	566	C
Maranoa Regional	0.79	2006	36.87	0.00	5	0.00	4335	235	C
Townsville City	9.66	2012	33.77	0.00	8	0.04	184	9146	C
Balonne Shire	0.78	2002	31.57	0.00	5	0.00	1501	713	C
Ipswich City	0.00	2013	25.45	0.00	3	0.00	13,132	93	H
South Burnett Regional	0.00	2013	25.28	0.00	2	0.00	1768	39	H
Gold Coast City	0.00	2003	24.88	0.00	3	0.00	10,879	116	H
Toowoomba Regional	0.00	2014	24.59	0.00	3	0.00	9872	65	H
Gympie Regional	0.00	2002	24.18	0.00	3	0.00	8600	142	H
Ipswich City	0.00	2010	23.86	0.00	2	0.00	400,000	72	H
South Burnett Regional	0.00	2017	23.58	0.00	3	0.00	7053	88	H
Murweh Shire	0.36	2017	23.15	0.00	3	0.00	6106	100	C
Barcaldine Regional	7.08	2015	22.91	0.00	4	0.00	835	1025	C
Southern Downs Regional	0.00	2015	22.58	0.00	3	0.00	5046	123	H
Toowoomba Regional	0.00	2015	19.67	0.01	2	0.00	50,862	16	H
Southern Downs Regional	0.00	2009	18.70	0.02	2	0.00	31,299	26	H
Southern Downs Regional	0.00	2003	18.11	0.02	2	0.00	23,251	35	H
Gympie Regional	6.12	2002	17.30	0.05	3	0.00	868	715	C

LLR: log-likelihood ratio; RR: relative risk; H: household cluster; C: community cluster.

**Table 3 pathogens-11-00830-t003:** Differences between Q fever cases within household and community clusters, and those outside clusters, in the proportions of reported exposures 1 month prior to disease onset.

Reported Exposure 1 Month Prior to Disease Onset	Community and Household Clusters vs. Cases Outside a Cluster				Household Clusters vs. Cases Outside a Cluster				Community Clusters vs. Cases Outside a Cluster			
	Answer from Cases that Belongs to Household and Community Cluster		Chi-Square Statistic	*p*-Value	Answer from Cases that Belongs to Household Cluster		Chi-Square Statistic	*p*-Value	Answer from Cases that Belongs to a Community Cluster		Chi-Square Statistic	*p*-Value
	Yes	No			Yes	No			Yes	No		
	*n* (%)	*n* (%)			*n* (%)	*n* (%)			*n* (%)	*n* (%)		
Aware of Q fever vaccination	50 (42.7)	67 (57.3)	0.08	0.77	42 (47.7)	46 (52.3)	0.28	0.60	15 (34.9)	28 (65.1)	1.27	0.26
Abattoir exposure	10 (8)	115 (92)	0.69	0.40	8 (8.2)	89 (91.8)	0.37	0.54	6 (14.3)	36 (85.7)	0.29	0.59
Work inside abattoir	7 (8.3)	77 (91.7)	0.91	0.34	5 (8.1)	57 (91.9)	0.68	0.41	5 (14.3)	30 (85.7)	0.01	0.91
Work on abattoir grounds	1 (2.6)	37 (97.4)	2.55	0.11	0 (0)	31 (100)	3.41	0.06	1 (8.3)	11 (91.7)	0.00	1.00
Visitor to abattoir	3 (8.8)	31 (91.2)	0.00	1.00	3 (10.3)	26 (89.7)	0.00	0.96	2 (20)	8 (80)	0.58	0.45
Assist/observe animal birth	61 (48.4)	65 (51.6)	73.28	<0.001	57 (58.8)	40 (41.2)	101.50	<0.001	11 (25)	33 (75)	0.62	0.43
Skinning/meat processing, etc.	16 (13.1)	106 (86.9)	0.36	0.55	12 (13)	80 (87)	0.25	0.62	9 (20.5)	35 (79.5)	0.53	0.47
Shooting/hunting	15 (12.2)	108 (87.8)	0.02	0.89	11 (11.8)	82 (88.2)	0.00	1.00	6 (13.3)	39 (86.7)	0.03	0.86
Work with wool	8 (6.4)	117 (93.6)	0.12	0.73	7 (7.3)	89 (92.7)	0.42	0.52	4 (9.1)	40 (90.9)	0.61	0.43
Work in shearing shed	8 (6.8)	109 (93.2)	0.33	0.57	5 (5.7)	83 (94.3)	0.00	1.00	6 (14)	37 (86)	5.02	0.03
Work in wool processing	5 (4.3)	110 (95.7)	0.53	0.47	3 (3.4)	84 (96.6)	0.00	0.98	4 (9.5)	38 (90.5)	4.76	0.03
Work with straw animal bedding	30 (24.4)	93 (75.6)	2.95	0.09	26 (27.7)	68 (72.3)	5.30	0.02	8 (18.2)	36 (81.8)	0.00	1.00
Work with animal manure, etc.	45 (36.6)	78 (63.4)	2.64	0.10	37 (39.4)	57 (60.6)	3.94	0.05	15 (34.1)	29 (65.9)	0.22	0.64
Attend saleyard/animal show	15 (12.1)	109 (87.9)	0.00	1.00	12 (12.8)	82 (87.2)	0.01	0.92	3 (6.7)	42 (93.3)	0.76	0.38
Live on farm	66 (52.8)	59 (47.2)	13.18	<0.001	58 (59.8)	39 (40.2)	21.29	<0.001	12 (28.6)	30 (71.4)	1.07	0.30
Visit farm	44 (44.9)	54 (55.1)	0.62	0.43	37 (50.7)	36 (49.3)	2.82	0.09	14 (35.9)	25 (64.1)	0.20	0.65
Exposed to livestock transport	59 (47.2)	66 (52.8)	0.80	0.37	40 (41.7)	56 (58.3)	0.03	0.87	25 (56.8)	19 (43.2)	2.98	0.08
Launder clothes of animal worker	44 (35.2)	81 (64.8)	12.13	<0.001	39 (40.6)	57 (59.4)	18.58	<0.001	11 (25)	33 (75)	0.06	0.81
Contact with infected person	53 (43.8)	68 (56.2)	312.47	<0.001	51 (54.8)	42 (45.2)	393.34	<0.001	8 (18.6)	35 (81.4)	9.51	<0.001
Consume unpasteurised milk, etc.	6 (4.8)	119 (95.2)	0.33	0.56	4 (4.2)	91 (95.8)	0.47	0.49	2 (4.4)	43 (95.6)	0.06	0.81
Contact with untreated water	52 (41.9)	72 (58.1)	1.62	0.20	43 (45.3)	52 (54.7)	3.15	0.08	11 (25)	33 (75)	2.02	0.16
Exposure to paddock dust, etc.	87 (72.5)	33 (27.5)	5.17	0.02	69 (73.4)	25 (26.6)	4.65	0.03	25 (62.5)	15 (37.5)	0.00	1.00
Live/work within 300 m of bush, etc.	91 (75.2)	30 (24.8)	8.50	<0.001	69 (73.4)	25 (26.6)	4.60	0.03	31 (75.6)	10 (24.4)	2.53	0.11

All reported exposure were analysed based on yes vs. no; community and household clusters (*n* = 221); household clusters (*n* = 146); community clusters (*n* = 75); total reported cases included in the analysis = 2175.

**Table 4 pathogens-11-00830-t004:** Summary of the Generalised Additive Model for the type of exposure reported between Q fever cases belonging to a cluster, and cases outside clusters.

Type of Exposure	Community and Household Clusters vs. Cases Outside a Cluster		
	Chi-Square Statistic	*p*-Value	Odd Ratio (2.5–97.5%)
Abattoir exposure	0.00	0.601	1 (0.998–1.002)
Assist/observe animal birth	4.40	0.036	3.17 (0.889–10.141)
Work with wool	0.71	0.315	0.04 (0–2.089)
Live on farm	0.00	0.626	1 (0.998–1.002)
Launder clothes of animal worker	1.82	0.100	1.93 (0.695–4.986)
Work on abattoir grounds	0.79	0.319	0 (0–0.624)
Work with animal manure, etc.	0.00	0.920	1 (0.999–1.001)
Visit farm	0.00	0.908	1 (0.997–1.003)
Contact with infected person	30.18	<0.001	39.11 (9.836–183.989)
Exposure to paddock dust, etc.	0.00	0.489	1 (0.999–1.002)
Visitor to abattoir	0.00	0.919	1 (0.997–1.003)
Shooting/hunting	0.00	0.525	1 (0.996–1.005)
Attend saleyard/animal show	3.65	0.078	2.62 (0.669–8.591)
Exposed to livestock transport	0.00	0.488	1 (0.998–1.002)
Consume unpasteurised milk, etc.	0.00	0.359	1 (0.979–1.023)
Live/work within 300 m of bush, etc.	0.31	0.250	0.86 (0.506–1.434)
Contact with untreated water	0.00	0.380	1 (0.995–1.005)

## Data Availability

The data supporting the findings of the article are not available publicly due to ethical reasons.

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
