# Peer review of "Profiling Risk Factors for Household and Community Spatiotemporal Clusters of Q Fever Notifications in Queensland between 2002 and 2017"

_pathogens, 2022, doi:10.3390/pathogens11080830_

Round 1
Reviewer 1 Report
The authors have studied the risk factors associated with Q fever in Queensland. The study is interesting but, in my opinion needs to be improved before becoming suitable for publication.
Main comments:
My main concern is that the results have been mainly focused on incidences and some useful data are not presented and/or discussed in a proper way. In particular, results presented at table 1. Here are some suggestions:
- The finding that there are differences between cases within clusters and outside the clusters with regard to laundering animal clothes. What impact does such finding have regarding epidemiology and prevention.
- The epidemiological and preventive impact of the finding that within and outside clusters differs in exposure to paddock dust. This is interesting to discuss in term of infection with fresh bacteria vs spore like particles.
- Similar to above discuss the significance of the difference between clusters with regard to living or working within 300 m bush.
- The authors mentioned that the Townville cluster was the biggest cluster with radius of 9.6 km and they suggest the role of the wind. It will be useful if the authors can add data about the wind and test the correlation between the wind and different cluster radius.
- Would the author consider testing the data using other models such as SAR. ref: Beyond Moran's I: Testing for Spatial Dependence Based on the Spatial Autoregressive Model - Li - 2007 - Geographical Analysis - Wiley Online Library.
Minor issues.
- Figure 1 needs to be improved.
- Error! reference source is not found: this is found in more than one place in the text, please correct.
- check also for minor grammatical errors and typos
Author Response
We would like to thank the reviewer for the comments and feedback on our manuscript. Below our answer and how we address the comments.
My main concern is that the results have been mainly focused on incidences and some useful data are not presented and/or discussed in a proper way. In particular, results presented at table 1. Here are some suggestions:
- The finding that there are differences between cases within clusters and outside the clusters with regard to laundering animal clothes. What impact does such finding have regarding epidemiology and prevention.
Thanks to the reviewer for providing comments and feedback to improve our manuscript. For the results related to differences between cases within clusters and outside clusters concerning laundering animal clothes, we improve this by expanding our discussion. The following paragraph has been added to the discussion to address the comment “The univariable model also shows those cases reporting contact with clothes worn by someone who work with animals were more likely to belong to a cluster. This type of exposure has been previously reported in a small outbreak, with three laundry workers infected with Q fever [22]. Our results suggest that laundered clothes from animal workers are a potential risk source for Q fever clusters”
- The epidemiological and preventive impact of the finding that within and outside clusters differs in exposure to paddock dust. This is interesting to discuss in term of infection with fresh bacteria vs spore like particles.
We have included the following paragraph in the discussion to address the comment “Similarly, we identified that notifications that reported exposure to paddock dust were more likely to belong to a community or household cluster. This result is consistent with the importance of aerosol transmission in Q fever infections [9] due to the capacity of the bacteria to survive in the environment, with viable bacteria being recovered from soil up to 20 days after inoculation [23]”
- Similar to above discuss the significance of the difference between clusters with regard to living or working within 300 m bush.
We have included the following paragraph in the discussion to address the comment “Cases reporting living or working within 300 m of bushland were also more likely to belong to a cluster. This may be an indicator that the environment is playing an important role in the maintenance of the bacteria that could drive the Q fever clusters”
- The authors mentioned that the Townville cluster was the biggest cluster with radius of 9.6 km and they suggest the role of the wind. It will be useful if the authors can add data about the wind and test the correlation between the wind and different cluster radius.
The association between Q fever and wind has been described before and reviewed by Clark and Soares-Magalhaes 2018. Moreover, the nature of the geographical unit of the spatiotemporal analysis may have affected the size of the clusters in our study. As we used the centroid of each mesh-block for the analysis, the size of the clusters may be affected by the average size of each mesh-block. For example, the average size of a mesh-block in Townsville is 22 km2 while the average all the whole Brisbane area is 0.1 km2. Another fact that it needs to be considered is the sparsity of the data, with not many cases in the same area. To understand better the relation between wind and Q fever clusters, we currently have a field team working on airborne and collecting air and dust samples from hotspots and the results of that study will become available.
To clarify that the size of Townsville clusters may also be affected by the geographical unit we included the following sentence in the discussion. “However, other factors such as the average size of the mesh-block in Townsville area (larger than mesh-blocks in southeast regions) could have an effect in the size of the clusters”
- Would the author consider testing the data using other models such as SAR. ref: Beyond Moran's I: Testing for Spatial Dependence Based on the Spatial Autoregressive Model - Li - 2007 - Geographical Analysis - Wiley Online Library.
Thank you to the reviewer for the suggestion. We discussed the possibility to use SAR model as the reviewer suggested, however, we concluded that for this specific study, we aimed to profile the communities and household Q fever clusters, focusing only on the presence or not of a cluster. The SAR model focuses on interpolating risk between locations, which is beyond the scope of this study.
Minor issues.
- Figure 1 needs to be improved.
We apologise for this. The figure had a lower resolution than the word document. We have improved the quality of figure 1 within the word document and we have also included the figure as a separate file at a higher resolution.
- Error! reference source is not found: this is found in more than one place in the text, please correct.
We have corrected these errors by manually updating all tables and figures.
- check also for minor grammatical errors and typos
Grammatical and typos have been checked and some sections of the manuscript have been reworded.
Reviewer 2 Report
The spatial distribution of Q fever is always interesting to learn more about it. But the presented study moves away from the year 2022. Taking into account the title of the article, the authors have not reflected the importance, nor have they made a translation of the results to the current date, since in fact the authors reflect "In conclusion, our results demonstrate that clusters of Q fever notifications are temporally stable and geographically circumscribed indicating a persistent common exposure. Perhaps the authors should consider changing the title of the article”.
In addition, references are missing in some lines of the results section.
Table of supplementary materials, it would be necessary to add abbreviated information that appears in the table
With respect to the presentation of the statistical results and the discussion of them is appropriate for the time interval analyzed. If the objective of the study is modified when considering that 6 years have passed, the conclusion should be modified.
Author Response
Please find attached our response to the reviewer's comments.
The spatial distribution of Q fever is always interesting to learn more about it. But the presented study moves away from the year 2022. Taking into account the title of the article, the authors have not reflected the importance, nor have they made a translation of the results to the current date, since in fact the authors reflect "In conclusion, our results demonstrate that clusters of Q fever notifications are temporally stable and geographically circumscribed indicating a persistent common exposure. Perhaps the authors should consider changing the title of the article”.
- In addition, references are missing in some lines of the results section.
We have corrected these errors by manually updating all tables and figures.
- Table of supplementary materials, it would be necessary to add abbreviated information that appears in the table
Thank you to the reviewer for pointing out this issue. We have improved the table in the supplementary material by including the abbreviation information.
- With respect to the presentation of the statistical results and the discussion of them is appropriate for the time interval analyzed. If the objective of the study is modified when considering that 6 years have passed, the conclusion should be modified.
We thank the reviewer for this comment, however, the situation of Q fever cases in Queensland, Australia, has not changed in the last 5 years, and therefore the result of this study reflects the current Q fever situation in the state. Even though we agree that having a more updated database would be more suitable, unfortunately, the validation of the data by Queensland Health, Public Health Act 2005, Notifiable Conditions System (NoCS) managed by the Communicable Disease Branch of Queensland Health, can take approximately two years. On top of that, human ethics approval and data custodian approval are required and is a process that can take another year or more.
Round 2
Reviewer 2 Report
-
The new version of the article "Profiling risk factors for household and community spatiotemporal clusters of Q fever notifications in Queensland between 2002 and 2017" contains the modifications previously proposed to the authors
Only note that the tables included in the manuscript must be renamed, since there are two tables 1 and two tables 2.
Author Response
The new version of the article "Profiling risk factors for household and community spatiotemporal clusters of Q fever notifications in Queensland between 2002 and 2017" contains the modifications previously proposed to the authors
Only note that the tables included in the manuscript must be renamed since there are two tables 1 and two tables 2.
We would like to thank the reviewer for the time and comment. Unfortunately, I am not sure what is the error in the tables. The tables in the manuscript are named as follows and I have included the line where each table is located within the manuscript. Please, apologise if I misunderstood the comment.
- Table 1, in line 113, has been named "Yearly spatial-temporal analysis on Q fever incidence in Queensland, 2002–2017"
- Table 2, in line 143, has been named" Significant Q fever household and community clusters using space-time analysis in Queensland between 2002 and 2017"
- Table 3. In line 178, "Differences between Q fever cases within household and community clusters and those outside clusters in the proportions of reported exposures 1 month prior to disease onset.
-
Table 4. in line 182. "Summary of the Generalised Additive Model for the type of exposure reported between Q fever cases belonging to a cluster and cases outside clusters"
- And Table S1 in the supplementary material, "Q fever clustering using space-time analysis in Queensland between 2002 and 2017
I hope that I was able to clarify the table's issues.
Kind regards,
Dr Tatiana Proboste